



# A new approach to estimate ice dynamic rates using satellite observations in East Antarctica

Bianca Kallenberg[1], Paul Tregoning[1], Janosch F. Hoffmann[2], Rhys Hawkins[1], Anthony Purcell[1], Sébastien Allgeyer[1]

[1]Research School of Earth Sciences, Australian National University, Canberra, ACT, 0200, Australia
[2]College of Computer, Mathematical, and Natural Science, University of Maryland, College Park, MD 20742, USA

*Correspondence to*: Bianca Kallenberg (bianca.kallenberg@anu.edu.au)

**Abstract.**

Mass balance changes of the Antarctic ice sheet are of significant interest due to its sensitivity to climatic changes and its contribution to changes in global sea level. While regional climate models successfully estimate mass input due to snowfall, it remains difficult to estimate the amount of mass loss due to ice dynamic processes. It's often been assumed that changes in ice dynamic rates only need to be considered when assessing long term ice sheet mass balance; however, two decades of satellite altimetry observations reveal that the Antarctic ice sheet changes unexpectedly and much more dynamically than previously expected. Despite available estimates on ice dynamic rates obtained from radar altimetry, information about changes in ice dynamic rates are still limited, especially in East Antarctica. Without understanding ice dynamic rates it is not possible to properly assess changes in ice sheet mass balance, surface elevation or to develop ice sheet models. In this study we investigate the possibility of estimating ice dynamic rates by removing modelled rates of surface mass balance, firn compaction and bedrock uplift from satellite altimetry and gravity observations. With similar rates of ice discharge acquired from two different satellite missions we show that it is possible to obtain an approximation of ice dynamic rates by combining altimetry and gravity observations. Thus, surface elevation changes due to surface mass balance, firn compaction and ice dynamic rates can be modelled and correlate with observed elevation changes from satellite altimetry.





## 1 Introduction

Assessing and understanding ice mass balance of the Antarctic Ice Sheet (AIS) is challenging due to the remoteness and extensive ice cover of the continent, resulting in a sparse network of field observations to provide information about the climate, mass balance or bedrock uplift rates. In order for an ice sheet

to be in balance the amount of ice lost, due to the dynamic processes of meltwater runoff and solid ice discharge over the grounding line, needs to be balanced by accumulated snowfall. If one exceeds the other, the ice sheet either gains or loses mass, resulting in a change in ice sheet mass balance (Cuffey and Paterson, 2010). The surface processes of snowfall, snowmelt and subsequent runoff, sublimation, evaporation and snowdrift add, remove or distribute snow and define the surface mass balance (SMB)

(e.g. Lenaerts et al., 2012; Van Wessem et al., 2014). Changes in SMB occur primarily in the firn layer that covers the AIS, the intermediate product between snow and ice (Ligtenberg et al., 2011). Temperature variations, overburden pressure, deformation and repositioning of snow grains causes snow to densify until it reached the density of glacier ice ($\sim$917 kg m$^{-3}$) (Herron and Langway, 1980; Ligtenberg et al., 2011). This results in an elevation change without changing the mass of the ice sheet.

When thoroughly evaluated with field observations and potentially downscaled using statistical interpolation methods, regional climate models can be used to simulate fields of SMB components, temperature and near-surface wind speed. Ice loss rates can be obtained by combining individual estimates of accumulation, ablation and dynamic ice loss, with the difference between mass input and mass output providing the mass balance of the ice sheet. While SMB can be taken from regional climate

models, estimates on ice discharge are limited and difficult to obtain. Satellite radar altimetry is used to retrieve information about ice velocity rates and ice thickness from airborne radar or, in the absence of direct observations, a floatation criterion provides the amount of ice that discharges across the grounding-line into the ocean (Rignot and Thomas, 2002; Rignot et al., 2008; Allison et al., 2009). Commonly, ice dynamics rates are either taken from these by satellite altimetry derived estimates

(Shepherd et al., 2012; Sasgen et al., 2013), or assumed to be insignificant when studying short-term changes (e.g. Ligtenberg et al., 2011). However, unexpected changes in ice sheet dynamic have been observed in the past decades, with some glaciers found to accelerate, while others decelerated (Rémy and Frezzotti, 2006). In general, ice dynamics are not well known and information about ice dynamic





rates are limited (Rignot, 2006; Rignot et al., 2008). This becomes an issue when assessing ice mass balance and surface elevation changes, or establishing ice sheet models.

Although satellite observations help provide information about temporal and spatial changes in ice mass and ice volume, large uncertainties remain when interpreting the signals and assigning the origin of

change. Ice mass balance can be measured directly from gravity observations but needs to be separated into the possible changes caused by SMB, ice dynamics and Glacial Isostatic Adjustment (GIA), the response of the lithosphere adjusting to changes in surface loads. Changes in ice sheet thickness can be obtained from altimetry observations but need to be separated into the change caused by SMB, ice dynamics, GIA and/or firn compaction. Observed elevation changes can subsequently be converted to

changes in mass, by employing firn densities.

In this study we obtain an estimate of ice dynamic rates by combining modelled SMB rates using the Regional Atmospheric Climate MOdel (RACMO2), Gravity Recovery And Climate Experiment (GRACE) and laser altimetry observations from the Ice, Cloud and land Elevation Satellite (ICESat). We combined our  modelled ice dynamics rates with elevation changes due to SMB and

firn compaction, we used to model surface elevation changes.  We then compared these changes to direct observations of ice surface height from ICESat. A study site in East Antarctica has been chosen due to the increase in mass that has been observed there by GRACE and altimetry, suggesting a thickening of the ice sheet. We found that estimated ice dynamic rates obtained from GRACE and ICESat are of similar magnitude and can be used to model surface elevation changes

that are comparable with altimetry observations.

## 2 Study area

The chosen study area combines Enderby Land, Kemp Land and MacRobertson Land, and parts of Dronning Maud Land and Princess Elizabeth Land (hereafter referred to as Enderby Land for simplicity), recording a general positive mass trend across this region (e.g. Shepherd et al., 2012;

Sasgen et al., 2013).

The study area is assumed to be a stable region (e.g. Rignot et al., 2008), with the ice sheet predominantly located on bedrock above sea level, making it less vulnerable to changes in ocean





temperatures. The major outlet glaciers of this region are the Lambert and Mellor glaciers feeding the Amery Ice Shelf in the east, together with the smaller (~3000 km$^2$) Fisher, Scylla and American Highland Glaciers. Only smaller glaciers are found along the remaining coastal region of Enderby Land, including the Shirase, Rayner, Thyer and Robert glaciers (Fig. 1). Previous research

found the ice sheet to be largely in balance across this area, possibly even slightly thickening (Rignot, 2006; Rignot et al., 2008; Rignot et al., 2013).

**3 Data sets and methods**

We use observational measurements of mass variations from the Gravity Recovery And Climate Experiment (GRACE) and surface elevation changes observed by laser altimetry using the Ice,

Cloud and land Elevation Satellite (ICESat). Both datasets are described in detail in Appendix A1.1 and A1.2, respectively, together with the employed climate model RACMO2/ANT (Appendix A1.3) that is used to model the trend in SMB and to force the firn compaction model (Appendix A1.4). Two versions of the RACMO2 model are used here, RACMO2.1 and RACMO2.3.

The SMB used throughout this paper is the sum of snowfall, evaporation/sublimation, snowdrift

and runoff. The SMB components are provided in kg m$^{-2}$ t$^{-1}$, where t is the temporal resolution of the model (Appendix A1.3). In order to convert to a rate of snow equivalent (m t$^{-1}$) the SMB is divided by the surface snow density, determined individually for each grid point using the proposed parameterisation of Kaspers et al. (2004), together with a slope correction for Antarctica by Helsen et al. (2008):

$$\rho_s = -151.94 + 1.4266(73.6 + 1.06T + 0.0669A + 4.77W) \qquad (1)$$

where $T$ is the average annual temperature (in K), $A$ the average annual accumulation (in mm water equivalent (w.e.) yr$^{-1}$) and $W$ the average annual wind speed 10m above the surface (in m s$^{-1}$).

These values are typically determined from either observations or from numerical climate models such as RACMO2.





A change in surface elevation, dH/dt, as seen by satellite altimetry is caused by a combination of processes that affect ice sheet thickness, as well as the effect of GIA. The temporal change in surface height can be described as:

$$\frac{dH^{ICESat}}{dt} = \frac{dH^{SMB}}{dt} + \frac{dH^{fc}}{dt} + \frac{dH^{ice}}{dt} + \frac{dH^{GIA}}{dt} \qquad (2)$$

with the individual components representing elevation changes related to SMB ($dH^{SMB}/dt$), firn compaction ($dH^{fc}/dt$), ice dynamics ($dH^{ice}/dt$), and the elastic and viscoelastic response of the lithosphere combined under the term of GIA ($dH^{GIA}/dt$). While the process of firn compaction plays an important role in surface elevation changes it does not affect the overall mass balance of the ice sheet. Therefore, the general change in ice mass as detected by GRACE can be expressed as:

$$\frac{dM^{GRACE}}{dt} = \frac{dM^{SMB}}{dt} + \frac{dM^{ice}}{dt} + \frac{dM^{GIA}}{dt} \qquad (3)$$

with the individual components representing a change in mass due to SMB ($dM^{SMB}/dt$), ice dynamics ($dM^{ice}/dt$), and GIA ($dM^{GIA}/dt$).

With the components that assemble $dM^{SMB}/dt$ being represented by regional climate models simulating near surface climate in Antarctica, and $dM^{GIA}/dt$ modelled by available GIA models, $dM^{ice}/dt$ remains the only unknown in Equation 3. Therefore, an estimate of $dM^{ice}/dt$ can be obtained by removing $dM^{SMB}/dt$ and $dM^{GIA}/dt$ from the GRACE observations:

$$\frac{dM^{ice}}{dt} = \frac{dM^{GRACE}}{dt} - \frac{dM^{SMB}}{dt} - \frac{dM^{GIA}}{dt} \quad . \qquad (4)$$





Similarly, the same approach can be used to obtain dH$^{ice}$/dt from altimetry:

$$\frac{dH^{ice}}{dt} = \frac{dH^{ICESat}}{dt} - \frac{dH^{SMB}}{dt} - \frac{dH^{fc}}{dt} - \frac{dH^{GIA}}{dt} \quad .$$

(5)

This leaves us with estimated ice dynamic rates represented as a change in ice mass, $\frac{dM^{ice}_{GRACE}}{dt}$, and

surface elevation, $\frac{dH^{ice}_{ICESat}}{dt}$. We can convert to/from mass rate and height rate by

dividing/multiplying by the density of glacier ice. Thus, observations from each satellite mission
can provide an independent estimate of the ice dynamics.

We first correct both observational measurements, GRACE and ICESat, for GIA using three

available GIA models: the W12a model of Whitehouse et al. (2012), the ICE-6G_C (VM5a) model of
Peltier et al. (2015) and the recomputed version ICE6G_ANU of Purcell et al. (2016). Changes due
to SMB are modelled using RACMO2.3/ANT, and the total trend in SMB, for the period 2003-2009,
is obtained using the monthly SMB (kg m$^{-2}$ mth$^{-1}$). The change in dH$^{SMB}$/dt is acquired by dividing
dM$^{SMB}$/dt by the density of surface snow (Eq.1), and the rate of change due to firn compaction,

dH$^{fc}$/dt, is taken into account by using our modelled firn compaction rates. Each month, the total
SMB is computed and a monthly average firn compaction rate is removed from the SMB, before

calculating the overall trend dH$^{SMB}$/dt over 2003-2009. Finally, the obtained $\frac{dH^{ice}_{ICESat}}{dt}$ rates can be

converted to $\frac{dM^{ice}_{ICESat}}{dt}$ by multiplying by the density of glacier ice ($\sim$ 917 kg m$^{-3}$), while the $\frac{dM^{ice}_{GRACE}}{dt}$

rates are converted to $\frac{dH^{ice}_{GRACE}}{dt}$ by dividing by the density of glacier ice.

If ICESat and GRACE detect the same signal, the obtained $\frac{dM^{ice}_{ICESat}}{dt}$ estimates should correlate with

$\frac{dM^{ice}_{GRACE}}{dt}$ and vice versa, $\frac{dH^{ice}_{ICESat}}{dt}$ with $\frac{dH^{ice}_{GRACE}}{dt}$. Moreover, modelling surface elevation changes

($\frac{dH^{Mod}}{dt}$) found by removing $\frac{dH^{ice}_{GRACE}}{dt}$ from the modelled dH$^{SMB}$/dt and dH$^{fc}$/dt estimates should

approximate the ICESat observations:





$$\frac{dH^{Mod}}{dt} = \left( \frac{dH^{SMB}}{dt} - \frac{dH^{fc}}{dt} \right) - \frac{dH^{ice}_{GRACE}}{dt}. \tag{6}$$

Conversely, $\frac{dH^{ice}_{ICESat}}{dt}$ not being equal to $\frac{dH^{ice}_{GRACE}}{dt}$ indicates that there must be an error in either the observations or the SMB model.

**4 Results and discussion**

The chosen region is part of a vast area in East Antarctica that shows an increase in mass, suggesting that the ice sheet is growing in this region. The signal the GRACE satellites detect includes changes in mass due to accumulation, ice discharge and GIA. In Figure 2 we show the observed change in mass measured by GRACE, on a spatial scale (Fig. 2a) for our study period,

and locally for a chosen location near 67°S 54°E (Fig. 2b) for the entire operational period. In order to obtain the signal that is solely due to ice mass changes the contribution of GIA needs to be removed. In Figure 3 we show the GRACE signal corrected for GIA uplift rates using the ICE-6G_C (VM5) model by Peltier et al. (2015), W12a model by Whitehouse et al. (2012) and the recomputed version ICE6G_ANU of Purcell et al. (2016), respectively. Using ICE-6G_C (VM5) (Fig.

3a) significantly reduces the observed positive anomaly in Enderby Land, while applying W12a (Fig. 3b) and ICE6G_ANU (Fig. 3c) results in a smaller reduction of the mass anomaly, yielding a similar corrected GRACE signal. Due to the similarity between the W12a and ICE6G_ANU model the W12a model was chosen to correct the satellite observations for GIA, although the effect on the ice dynamic rates is insignificant between the models. With the contribution of GIA removed,

the signal should only comprise contributions from snowfall and ice discharge, revealing a positive trend in SMB between 30°E and 70°E, with a positive anomaly of ~32 ± 8 mm w.e. yr$^{-1}$ near 40°E and 55°E, showing a significant increase in mass between 2003-2009 (Fig. 3b).

The modelled trend in SMB and surface elevation due to SMB and firn compaction can now be removed from the GRACE and ICESat observations (Eq. 7 and Eq. 9), to obtain $\frac{dM^{ice}_{GRACE}}{dt}$ and $\frac{dH^{ice}_{ICESat}}{dt}$





and, subsequently, $\frac{dH^{ice}_{GRACE}}{dt}$ and $\frac{dM^{ice}_{ICESat}}{dt}$ by dividing (multiplying) by the density of glacier ice. We converted the rate of change of surface elevation due to ice dynamic rates obtained from ICESat into spherical harmonics to be comparable with $\frac{dH^{ice}_{GRACE}}{dt}$. The ice dynamic rate estimates are shown in Figure 4, comparing estimates obtained using two different SMB models: RACMO2.1 and

RACMO2.3. For both RACMO2 models the ice dynamic estimates are of somewhat similar rate for the two estimates obtained from GRACE and ICESat, with the greatest ice dynamic rates obtained between 30°E and 50°E. Using RACMO2.3 the ice dynamic estimates are significant smaller and primarily present between 30°E and 60°E with an estimated rate of -0.08 to -0.13 m yr⁻¹. Using RACMO2.1 yields ice dynamic rates of -0.08 m yr⁻¹ and above along the entire ice sheet margin of

our study region. The differences between the ice dynamic rates obtained by GRACE and ICESat (Fig. 4c and 4f) show significant changes between the former and latter RACMO2 versions, with an RMS error of 0.019 m yr⁻¹ and 0.021 m yr⁻¹ for RACMO2.3 and RACMO2.1, respectively. Generally, RACMO2.3 shows a smaller difference between the obtained ice dynamic rates, improving results across the study area. However, regions remain that exhibit differences of ±

0.05 m yr⁻¹. In both $\frac{dH^{ice}_{ICESat}}{dt}$ rates a positive trend is estimated across the center of the region. This is the result of a slightly positive elevation trend that has been recorded by ICESat observations in region D (Fig. 5b).

Finally, the total change in surface elevation is modelled, based on dH$^{SMB}$/dt, dH$^{fc}$/dt and $\frac{dH^{ice}_{GRACE}}{dt}$ (Fig. 5a). Using RACMO2.3, the result of the modelled rate of change of surface elevation reveals a

similar pattern to the ICESat observations (Fig. 5b). In region A both the negative trend between 28°E and 32°E and the positive trend at 34°E is modelled. In region B a general negative trend is recorded along the ice margin with a positive trend near 46°E. Both signals appear in our modelled elevation trend, though at a smaller magnitude. Similarly for region C, which shows a general negative trend across the region, with the lowest trend near 51°E and a strong positive

signal at 56°E. While the general negative trend is obtained in the model, the strong negative signal near 51°E is not present. The strong positive signal at 56°E is modelled, although it appears





slightly over predicted, covering a larger region than seen in the ICESat observations. Across region D ICESat monitored an overall increase in elevation, especially near 70°E, together with a slight decrease in surface height along the margin between 58°E and 70°E and at the Mellor Glacier (Fig. 1) near 68°E. Similar to the ICESat observations the general positive trend across the
region is modelled, together with the positive signal near 70°E, as well as a slight negative trend across the margin. However, the strong negative trend at the Mellor Glacier is lacking, though the region shows a slight negative trend. Although the modelled trend in surface elevation suggests similar behaviour to the altimetry observations, the signal generally appears damped compared to the ICESat observations. This is likely caused by the loss of spatial resolution through the use of
degree 80 spherical harmonics (the resolution of the GRACE gravity fields) to remove the ice dynamic rates.

Uncertainties are estimated for the satellite observations and models, with the methods and individual uncertainty estimates described in Appendix A2. The uncertainty estimated for the modelled surface elevation trend varies between near zero and ~6 cm yr$^{-1}$ across the interior and
along large parts of the ice sheet margins, and up to 12 cm yr$^{-1}$ for the two locations with high SMB rates.

**5 Conclusion**

Ice dynamic rates can be estimated independently from GRACE and satellite altimetry observations through the removal of GIA signals, SMB and, in the case of altimetry, firn
compaction signals. Both approaches depend upon a separate SMB model, although in different ways since SMB causes a mass change in GRACE observations but a height change in altimetry observations. Therefore, any errors in the modelled SMB lead to differences in the ice dynamic rates derived from GRACE versus altimetry. Thus, this approach provides a new and independent means of assessing the accuracy of SMB models. We showed that the differences between the old
and new RACMO2 versions yield significantly different ice dynamic rate estimates, with RACMO2.3 producing smaller differences between the GRACE- and ICESat-derived estimates.





Although different GIA models affect GRACE and altimetry observations in different ways, changes in GIA models have a small effect on ice dynamic rates and so are not responsible for different estimates using the two satellite techniques. We believe that the differences are not based on errors in the ICESat observations as most of the greatest differences occur in regions where

ICESat uncertainties are low (Appendix A2), in particular the significant difference occurring inland within the study region. Moreover, modelling the rate of change of surface elevation based on ice dynamic rates obtained from GRACE observations and RACMO2.3 estimates positive and negative changes in elevation in the same regions as ICESat detects corresponding trends, though the rates appear slightly under-estimated compared to the altimetry observations. Therefore, it

appears that the dominant driver in the differences of the modelled ice dynamic rates and surface elevation trends are the changes of the SMB rates within the RACMO2 model, with RACMO2.3 providing a more accurately modelled rate of change of surface elevation. Thus ice dynamics rates derived from GRACE and altimetry observations not only provides information about this process but could also help to identify areas where the SMB models are deficient.

**Appendix A. Supplement to data set and models**

**A1. Data sets and models**

**A1.1 GRACE**

We use the monthly gravity field solutions CNES/GRGS RL03-v3, provided by the Groupe de Researches de Géodésie Spatiale (GRGS). The RL03 solutions have a spatial resolution of degree

and order 80 (Lemoine et al., 2013) and have been chosen due to the stabilisation process that is applied to reduce noise in form of North-South striping. This is achieved by regularising the inversion for spherical harmonic coefficients (Bruinsma et al., 2010).

Temporary changes in the Earth's gravity field can be related to changes in surface mass due to the distribution of mass, as well as the elastic and viscoelastic (GIA) response of the lithosphere,

the instantaneous and long term signal to changes in surface load (Wahr et al., 1998). We obtain





mass anomalies by applying the equations that relate mass changes to gravity changes (Wahr et al., 1998) to obtain the change in mass due to SMB:

$$U^{w.e.}(\theta,\lambda,t) = R \sum_{n=2}^{N} \sum_{m=0}^{n} P_{nm}(\cos\theta) \frac{2n+1}{1+k_n^{elast}} \left( \Delta C_{nm}(t)\cos m\lambda + \Delta S_{nm}(t)\sin m\lambda \right) \tag{A1}$$

and due to the viscoelastic deformation, or GIA:

$$U^{visco}(\theta,\lambda,t) = R \sum_{n=2}^{N} \sum_{m=0}^{n} P_{nm}(\cos\theta) \frac{h_n^{visco}}{k_n^{visco}} \left( \Delta C_{nm}(t)\cos m\lambda + \Delta S_{nm}(t)\sin m\lambda \right) \tag{A2}$$

where R is the Earth's radius, $P_{nm}$ are the fully normalised Legendre functions, $n$ and $m$ are degree and order of the spherical harmonic coefficients, $\theta$ and $\lambda$ are colatitude and longitude, and $\Delta C_{nm}$, $\Delta S_{nm}$ are the spherical harmonic coefficients, at time $t$, of the GRACE anomaly fields. $k_n$ and $h_n$ are the elastic Love loading numbers (e.g. Pagiatakis, 1990) and the ratio of viscoelastic Love loading numbers (Purcell et al., 2011), depending on the degree.

**A1.2. ICESat**

Various methods are used to estimate surface elevation changes from ICESat observations, using either along-track measurements or measurements directly taken from the crossover location. Due to perturbations in the orbit, deviations of the repeated ground track occur and it is necessary to determine the surface topography to correct elevation changes due to surface slope
rather than changes in ice mass. Different methods have been applied to obtain surface elevation changes, using either along-track observations or crossover measurements (e.g. Slobbe et al., 2008; Gunter et al., 2009; Pritchard et al., 2009; Sørensen et al., 2011; Ewert et al., 2012).

Here we use the estimated rate of change of ice sheet elevation obtained from a newly developed technique that combines both crossover and along-track observations (Hoffmann, 2016). The
method allows estimation of the local surface slope using a digital elevation model that has been



derived from gridded estimates of ice height at ICESat crossover points. Over a crossover grid, that geographically spans all campaign crossovers of a location, a static grid was created on which heights were interpolated at the epochs of all campaigns. The estimate of the elevation change over time is made by computing a weighted least-square regression of the height time series of

each grid node and then computing a weighted mean value for all grid nodes to derive the crossover height rate. This not only allows to assess height rates at one location over time, but also to evaluate a digital elevation model directly from the data, which is used to estimate the slope at crossovers (Hoffman, 2016).

The slope estimates at the crossovers are then interpolated to remove the surface slope from the

along-track measurements. Although the elevation change estimates from along-track measurements are naturally less precise than the rate estimates at crossovers, combining both methods significantly increases the accuracy of the slope correction, providing a measure to validate along-track estimates (Hoffman, 2016).

### A1.3. RACMO2/ANT

The RACMO2/ANT regional climate model, used to obtain SMB estimates, adopts the dynamical processes from the High Resolution Limited Area Model (HIRLAM) and the physical atmospheric processes from the European Centre for Medium-range Weather Forecasts (ECMWF) (Reijmer et al., 2005) and is forced by ERA-Interim reanalysis data at the lateral boundaries (e.g. Ligtenberg et al., 2011; Lenaerts and van den Broeke, 2012). The latest version, RACMO2.3 (Van Wessem et al.,

2014), extends available model data from 1979-2012 (RACMO2.1) to 1979-2015 (RACMO2.3) and improves the temporal resolution from 6-hourly (RACMO2.1) to 3-hourly (RACMO2.3) (Ligtenberg per. comm., 2016). The horizontal resolution is 27 km and the vertical resolution 40 levels. Individual SMB components are provided including snowfall, evaporation/sublimation, and snowmelt, as well as snowdrift in RACMO2.3. Over Antarctica RACMO2/ANT is coupled with a

multilayer snow model, which estimates meltwater percolation, refreezing and runoff, as well as surface albedo and snowdrift (Van Wessem et al., 2014). The update in the physical parameters of RACMO2.3 results in a general increase in precipitation over the grounded East Antarctic Ice



Sheet, evaluated using in-situ observations, ice-balance velocities and GRACE measurements and showing a general improvement of the SMB (Van Wessem et al., 2014).

### A1.4. Firn compaction

We developed a firn compaction model based on the firn densification model of Ligtenberg e al.
(2011), using near surface climate provided by RACMO2.1. It is a one-dimensional, time-dependent model that estimates density and temperature individually for each layer and at each time step in a vertical firn column. The firn densification model of Ligtenberg et al. (2011) adds new snowfall instantly to the current top layer until the layer thickness exceeds ~15 cm (Ligtenberg, pers. comm., 2016), at which time it is divided in two layers. The properties of each
layer are passed on to both layers. If a layer becomes too thin, due to compaction or surface melt, the layer is merged with the next layer and assigned the average properties of both layers. Our model has been simplified to improve the computational time. Rather than adding new snowfall instantly to the top layer, we compute the monthly sum of SMB and use the monthly averaged surface temperature to estimate the densification rate, density and new temperature to obtain the
vertical velocity of the surface due to monthly firn compaction.

The model starts with a new firn layer created by the total SMB of one month and is built up by adding a new layer each month using monthly SMB values and mean surface temperatures. The surface snow density of each top layer is estimated (Eq. 3) and the densification rate is obtained using a dry snow densification expression proposed by Arthern et al. (2010):

$$\frac{d\rho}{dt} = CAg(\rho_i - \rho)e^{\left(\frac{-E_c}{RT} + \frac{-E_g}{RT_{av}}\right)} \tag{A3}$$

where C is the grain-growth constant (m s² kg⁻¹), independently calculated for densities below (C = 0.07) and above (C = 0.03) the critical density of 550 kg m⁻³, A is the accumulation rate (mm w.e.
yr⁻¹), g the gravitational acceleration, and $\rho$ and $\rho_i$ are the local density and the ice density (kg m⁻³), respectively. The exponential term includes the activation energy constants (kJ mol⁻¹) for creep




and for grain-growth, $E_c$ and $E_g$, respectively, the gas constant R (J mol$^{-1}$ K$^{-1}$) and the local temperature T, and annual average temperature $T_{av}$ (K).

The process of liquid water percolation and refreezing is incorporated as a function of snow porosity $P_s$ and density, as proposed by Coléou and Lesaffre (1998) (Ligtenberg et al., 2011; Kuipers Munneke et al., 2015):

$$L_W = 1.7 + 5.7 \left( \frac{P_S}{1 - P_S} \right) \tag{A4}$$

with the snow porosity:

$$P_S = 1 - \left( \frac{\rho}{\rho_i} \right) \tag{A5}$$

where $\rho$ is the density of the layer and $\rho_i$ the density of glacier ice.

The heat transport throughout the firn column is solved explicitly using the one-dimensional heat-transfer equation (Cuffey and Paterson, 2010):

$$\frac{dT}{dt} = \kappa \frac{d^2T}{dz^2} \tag{A6}$$

with the thermal diffusivity $\kappa$ and the depth $z$. Initially the heat-transfer equation consists of a term for heat conduction, advection and internal heating. However, initial heating is small within the firn layer and therefore neglected and the contribution of heat advection is taken into account by the downward motion of the ice flow (Cuffey and Paterson, 2010; Ligtenberg et al., 2011).

Finally, once the densification rate is estimated, the vertical velocity of the surface due to firn compaction, $V_{fc}$, can be assessed by integrating over the displacement of the compacted firn layers over the length of the firn column (Helsen et al., 2008):



$$V_{fc}(z,t) = \int_{z_i}^{z} \frac{1}{\rho(z)} \frac{d\rho(z)}{dt} dz \qquad\qquad (A7)$$

where z is depth, ρ density and dρ(z)/dt the densification rate.

Ligtenberg et al. (2011) found that Equation (A1) over-predicts the rate of densification for most regions in Antarctica, with the effect of the annual average accumulation being too large on the densification rate. They reintroduced an accumulation constant that previously had been proposed by Herron and Langway (1980) as α in $A^{\alpha}$ (below 550 kg $m^{-3}$) and β in $A^{\beta}$ (above 550 kg $m^{-3}$), initially chosen between 0.5 and 1.1 but later assumed to be α, β =1 (Zwally and Li, 2002;

Helsen et al., 2008). Ligtenberg et al. (2011) applied a modelled to observed ratio to correct for the accumulation dependence. We also found that Equation (A1) over-predicts the rate of densification, depending on the rate of the average annual accumulation.

However, due to our use of monthly layers, the ratio proposed by Ligtenberg et al. (2011) is no longer valid and we introduce new α and β, depending on the accumulation rate (Table A1). The

values for α and β represent a best fit and have been obtained by investigating different values across several model runs. This means that the firn compaction model is adjusted to fit available observations and is therefore assumed to be correct and invariant of SMB model changes.

In Figure A1a we show the average annual rate of firn compaction across the study site and in

Figure A1b the differences between our model and the model of Ligtenberg et al. (2011). Along the ice sheet margins and the Amery Ice Sheet our model overestimates their firn compaction rates by 5-10 cm $yr^{-1}$, while it underestimates rates by 7-12 cm $yr^{-1}$ in most other areas further inland, with up to 15 cm $yr^{-1}$ at two individual location near 28°E and between 68°E and 70°E. These differences are within our estimated uncertainty, based on the uncertainties provided for

the modelled SMB from RACMO2 (Appendix A3).



## A2. Uncertainty estimates

Uncertainties of the monthly GRACE solutions are derived following the method of Wahr et al. (2006) and are around 8 mm w.e. yr⁻¹ (Fig. A2a), reducing towards the Polar Regions due to denser ground track coverage (Wahr et al., 2006). The uncertainties of the ICESat observations are below 0.05 m yr⁻¹ in the interior, where a dense network of ground-tracks exists, and between 0.15 and 0.3 m yr⁻¹ along the ice sheet margins due to greater distances between the ground-tracks and steeper slopes along the margins (Hoffmann, 2016) (Fig. A2b).

For both RACMO2 models the overall uncertainty is given as 8% for the grounded ice sheet (Lenaerts et al., 2012; Van Wessem et al., 2014), resulting in an estimated uncertainty of less than 1 cm yr⁻¹ in the interior and up to 6 cm yr⁻¹ across the high SMB locations proposed in Enderby Land. The firn compaction model contains several error sources. In general, the complex physics of firn densification is still not fully understood, and the density of snow and firn is not well known, introducing large uncertainties into the computations (Sutterley et al., 2014). Error sources include the parameterisations to estimate surface snow density (Eq. 3) and the densification rate (Eq. A1), together with uncertainties within the forcing climate model RACMO2. Following the idea of Helsen et al. (2008) we obtain our error estimate for the firn compaction model by assessing the propagation of the major error sources that affect firn compaction rates. This was done by applying a bias to the accumulation (8%) and temperature (10 K (Reijmer et al., 2005; Maris et al., 2012)), as well as to the surface snow density (±20 kg m⁻³ (Helsen et al., 2008)). The propagation of the errors is calculated to obtain the total uncertainty of the firn compaction model (Fig. A2c). Across most of the study site the uncertainty is estimated to be around ±2-3 cm yr⁻¹. However, at the two locations with the high SMB rates the uncertainty is significantly larger and is estimated to be up to 8 cm yr⁻¹. To estimate the uncertainty of the modelled ice dynamic rates and modelled surface elevation change, the propagation of errors of the particular error source is obtained (Fig. A2d and A2e). Depending on the incorporated satellite mission the uncertainty for the modelled ice dynamic rates is up to 6 cm yr⁻¹ (GRACE, Fig. A2d) and up to 30 cm yr⁻¹ (ICESat, Fig. A2e), due to the larger error of the ICESat observations. The uncertainty of



the modelled elevation change is 0-12 cm yr$^{-1}$ (Fig. A2f), with the greatest error source being the firn compaction model.

### Acknowledgement

This work was supported in part by an Australian Antarctic Project grant (number AAP4160).

We would like to thank Stefan Ligtenberg for providing us with the RACMO2/ANT datasets and his extensive support and constructive help to establish our own firn compaction model.

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





| SMB (kg m$^{-2}$ yr$^{-1}$) | alpha | beta |
|---|---|---|
| <100 | 1.00 | 1.00 |
| 100-300 | 0.96 | 0.97 |
| 300-500 | 0.93 | 0.94 |
| 500-700 | 0.92 | 0.93 |
| 700-1000 | 0.90 | 0.86 |
| 1000-2500 | 0.88 | 0.86 |
| 2500-4000 | 0.87 | 0.84 |
| >4000 | 0.87 | 0.54 |

**Table A1:**

5 **Proposed values for the accumulation constants α and β used in our monthly firn compaction model. The constants are dependent on the accumulation rate and have been adapted to a best-fit.**





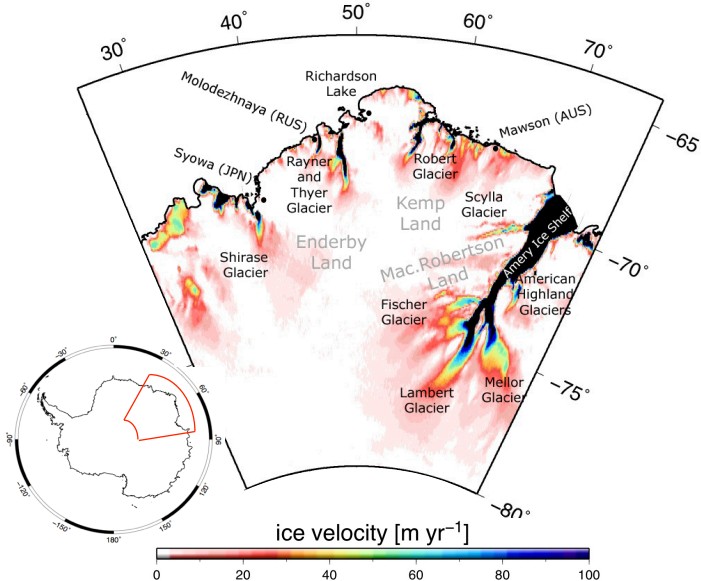

**Figure 1: Regional map of our study area including Enderby Land, Kemp Land and Mac.Robertson Land. The map includes the locations of permanent research stations and major outlet glaciers. Ice velocity rates are plotted, sourced from the NASA MEaSUREs program (Rignot et al., 2011; Mouginot et al., 2012), to identify glaciers and regions with dynamic ice loss.**



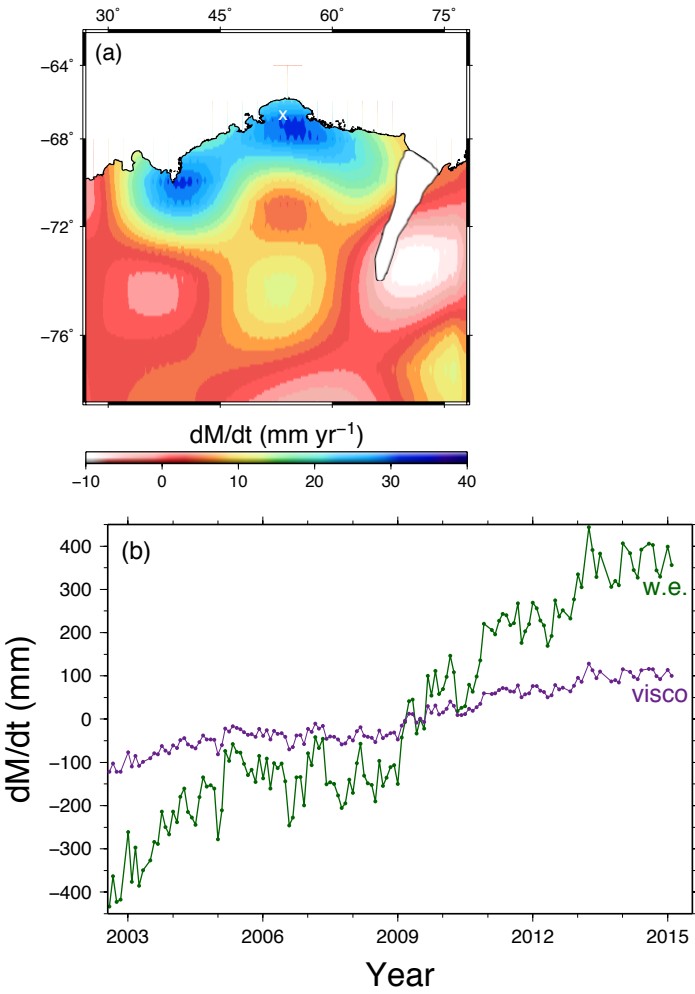

**Figure 2: (a) Trend of the observed mass anomalies in Enderby Land monitored by GRACE over the time span of 2003-2009, uncorrected for GIA. (b) The time series shows a change in gravity at a chosen location in Enderby Land (67S 54E) over the total observational period. The green line illustrates the change assuming the gravitational change is caused by a surface mass load and is expressed in water equivalent (w.e.) (Eq. A1 in Appendix), the purple line illustrates a change due to viscoelastic deformation (GIA) (Eq.A2 in Appendix).**





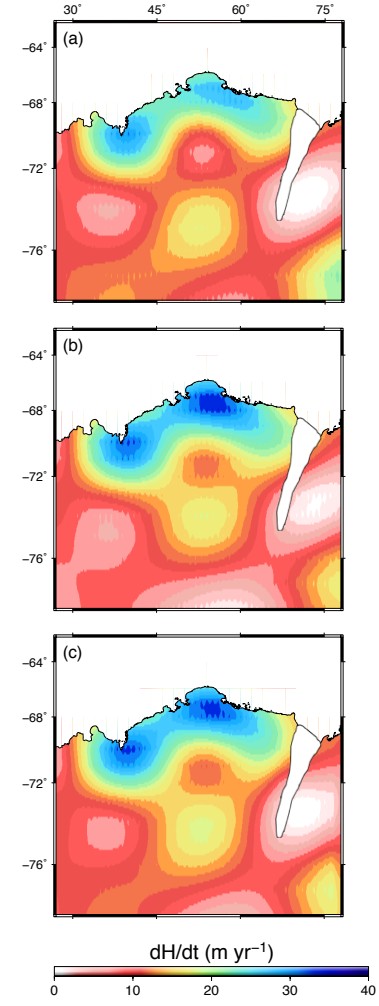

**Figure 3:** GRACE observations corrected for GIA uplift rates using (a) the ICE-6G_C(VM5) model by Peltier et al (2015), (b) the W12a model by Whitehouse et al. (2012), and (c) the ICE6G_ANU model by Purcell et al. (2016). The recomputed version ICE6G_ANU results in a similar correction to the W12a model.





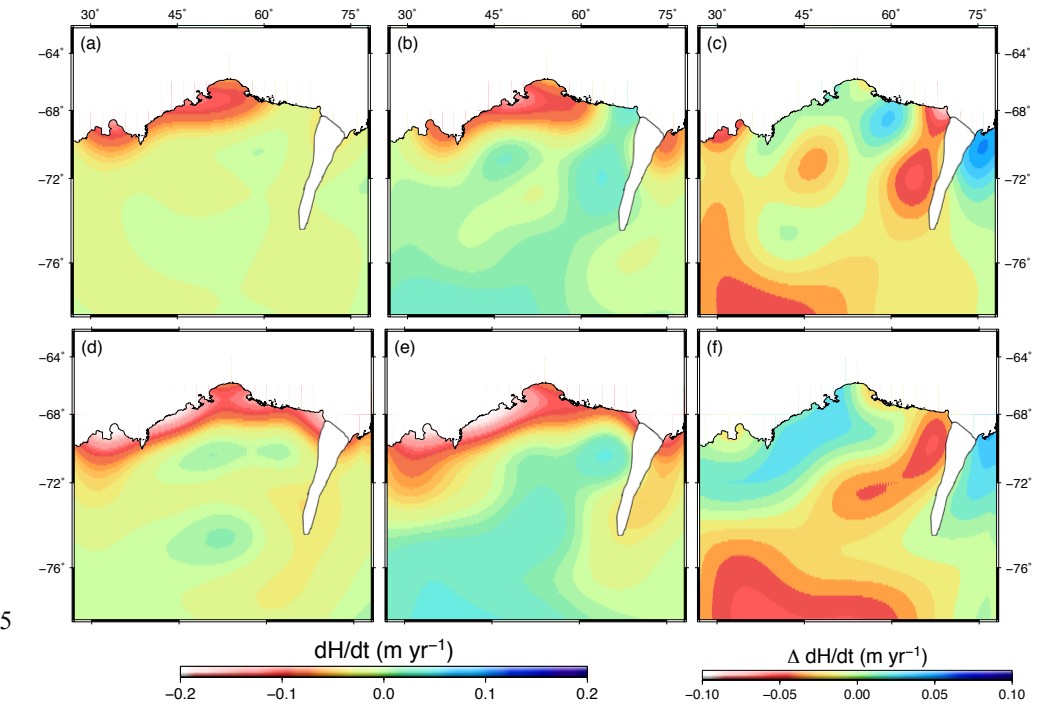

**Figure 4: Comparison between the modelled ice dynamic rates obtained from RACMO2.3 using (a) GRACE and (b) ICESat, and from RACMO2.1 using (d) GRACE and (e) ICESat. (c) and (f) show the difference between ice dynamic rates obtained from GRACE minus ice dynamic rates obtained from ICESat for RACMO2.3/ANT and RACMO2.1/ANT, respectively.**





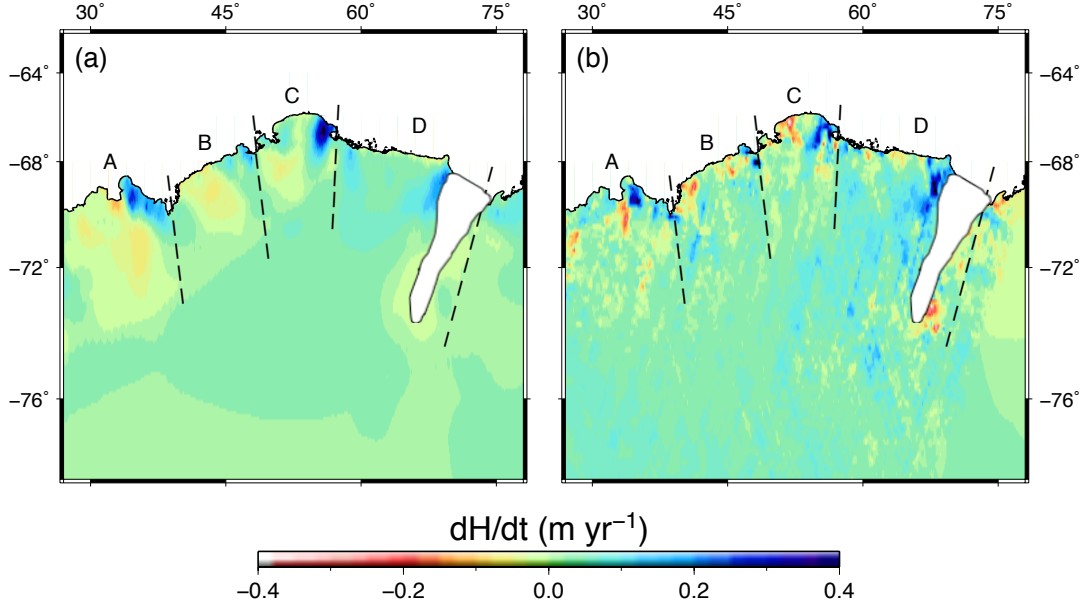

**dH/dt (m yr⁻¹)**

**Figure 5:** (a) our modelled rate of change of surface elevation retrieved by removing our estimated ice dynamic rates, obtained from GRACE, from the modelled trend in surface elevation (SMB-firn compaction) using RACMO2.3, compared to (b) the
5  ICESat observations.

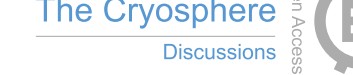



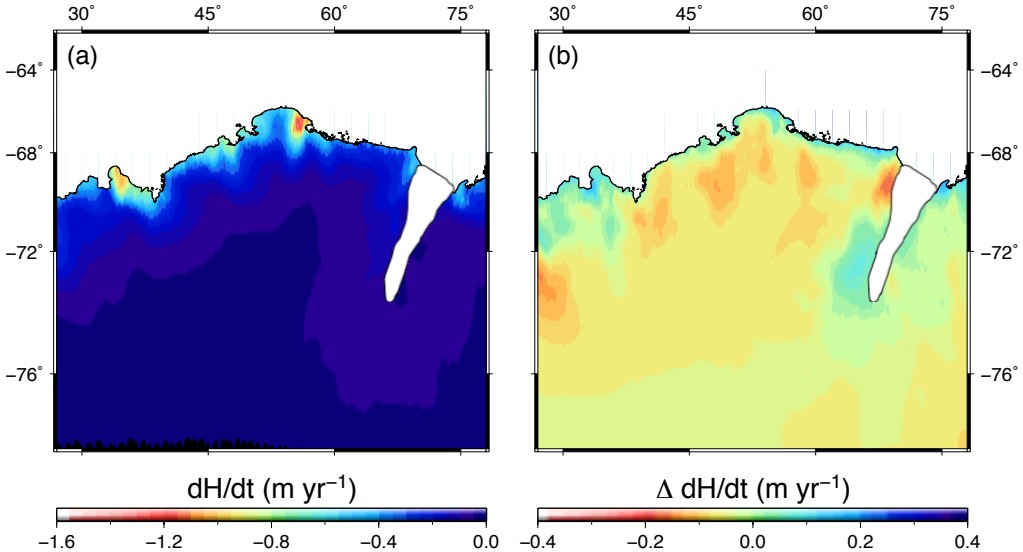

**Figure A1:** (a) Average annual vertical velocity rates due to firn compaction across the study site as obtained from our monthly firn compaction model, and (b) the differences between our model results and the firn densification model of Ligtenberg et al. (2011).





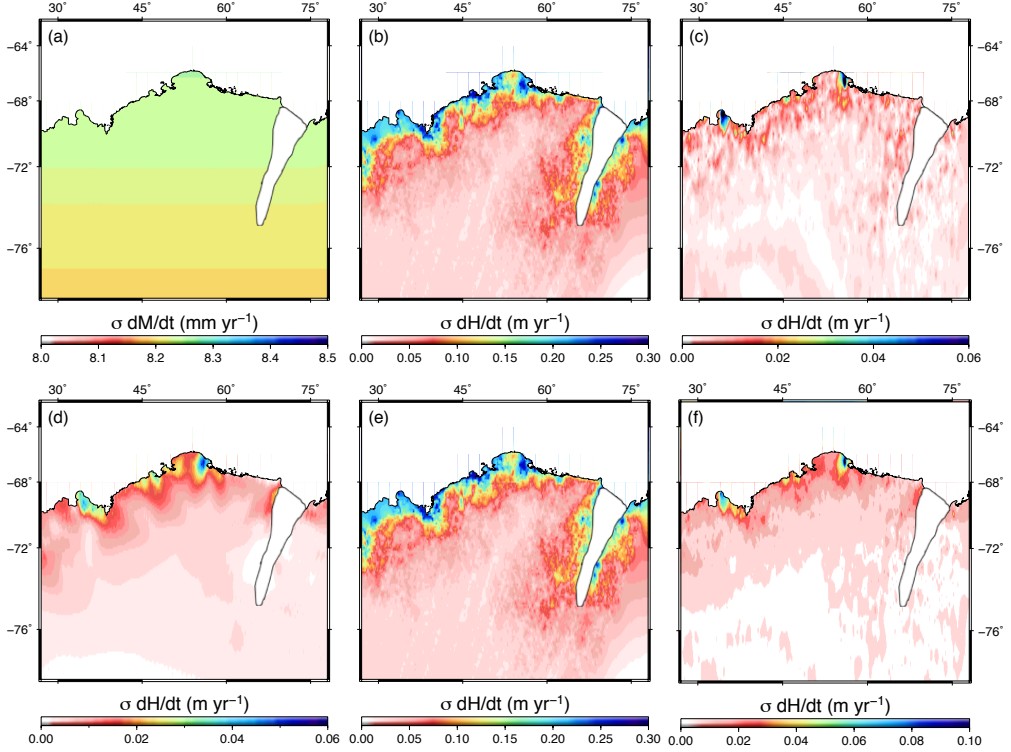

**Figure A2: Uncertainties estimated for (a) GRACE, (b) ICESat, (c) our monthly firn compaction model, ice dynamic rates using RACMO2.3 obtained from (d) GRACE, (e) ICESat, and the modelled surface elevation trend for (f) RACMO2.3. The greatest uncertainty comes from the ICESat measurements, with up to 30 cm yr⁻¹ at the margins, this results in greater uncertainties for the modelled ice dynamic rates obtained from the ICESat observations.**