# Peer review of "A new approach to estimate ice dynamic rates using satellite observations in East Antarctica"

_The Cryosphere, 2016_

## Referee Comment (RC1) · Anonymous Referee #1 · 31 Dec 2016

General Comment

Kallenberg et al report on a new approach for estimating ice dynamic rates for mass balance assessment from EO data and apply this for a study site in East Antarctica where an increase in ice mass has been observed in recent years. In the approach the ice dynamic are estimated by combining modelled SMB rates with gravity observations from GRACE and laser altimetry observations from ICESat. The derived IDR is combined with modelled elevation changes due to snow processes for comparison with measured elevation changes from ICESat. The authors find the estimated ice dynamic rates from GRACE and ICESat of similar magnitude and modelled elevation changes in correlation with direct altimetry observations. This is a well written, illustrated and

referenced methodological manuscript and a valuable and original contribution for the glaciology community. I would suggest a few minor corrections/clarifications to help improve the manuscript.

Specific Comments

Pg2 – Ln 20-23: There appears to be a mix up here. Ice velocity is derived from satellite radar interferometry and related methods in the mentioned studies, not from altimetry.

Pg5 – Ln 12-17: Studies have shown that leakage from oceanic geophysical signals may bias mass rate estimates for Antarctica from GRACE significantly. How is this quantified or dealt with in the approach?

Grammar, punctuation & style

Pg6 – Ln12: trend due to SMB

Pg7 – Ln24: Eq. 7 & Eq. 9 do not exist, I assume Eq. 4 & Eq. 5 are meant

Pg13 – Ln4: Ligtenberg et al.

Pg13 – Ln18: Eq. A3

Pg15 – Ln5&Ln11: Equation (A7)

Pg15 – Ln25: (Appendix A2)

Pg16 – Ln14: I assume Eq. 1 is meant here

Pg16 – Ln15: I assume Eq. A3 is meant here

Figures

Fig. 1: Invert color scale

Fig. 2: What is the white cross in Fig. 2a

Fig. 2: Y-axis label: dM/dt in mm or mm/yr?

Fig. 3: (m yr-1): please check scale here

[Figure]

---

## Referee Comment (RC2) · Anonymous Referee #2 · 4 Jan 2017

Summary: The manuscript back-calculates dynamic mass loss from East Antarctica using repeat GRACE and satellite altimetry observations paired with models of surface mass balance, firn compaction, and glacial isostatic adjustments. The authors find that rates of dynamic mass change inferred from the two different satellite observational platforms yield similar results, indicating that either platform can provide reasonable estimates of dynamic mass change when paired with current model outputs. The authors also infer that the good agreement between dynamic mass change estimates derived from the two different satellite platforms indicates that the most up-to-date RACMO model provides accurate estimates of surface mass balance for East Antarctica.

The results of the paper are interesting in that they show models of SMB, firn com-

paction, and GIA are accurate enough to allow us to tease-out the ice dynamics signal from repeat gravity and laser altimetry observations. There are several relatively small modifications that would improve the overall quality of the manuscript, including: 1) removal/adjustment of commas (described below), which often break sentences into somewhat awkward fragments, 2) incorporation of data and methods description that is currently contained in the appendices, and 3) change references to "ice dynamics rates" to "mass change rates due to dynamic change" or something similar My biggest concern is in regard to (2) above. I highly recommend that more information on the datasets and methods is included in the body text of the manuscript. The details regarding how (1) you process the GRACE and ICESat data, (2) the versions of RACMO used in the study differ from each other, and (3) the firn compaction converts SMB input to estimates of surface elevation change are incredibly important for assessing the validity of the results and for reproducing the method elsewhere. Although this will lengthen the manuscript, I think that adding more detail regarding the data used in the analysis is imperative. Given that the firn layer in Antarctica can be tens of meters thick and that the interpretation of altimetry data is incredibly sensitive to the accuracy of the firn compaction model (see Zwally et al., Journal of Glaciology, 2015 for an example) this is particularly important for the firn compaction model. As a follow-up, why use RACMO2.1 to run the firn compaction model when RACMO2.3 was found to produce better results for the GRACE-ICESat dynamic change comparison? This is inconsistent.

Detailed Comments: p. 2, line 4: Change to "and bedrock uplift rates."

p. 2, line 5: Move the comma after "ice lost" forward so it is after "balance"

p. 2, line 11: Either move the definition of firn so that it comes earlier in the sentence or remove it entirely. Currently it's in an awkward location.

p. 2, line 14: Change to "results in a change in the ice sheet surface elevation without…"

p. 2, line 15: Remove "potentially"

p. 2, lines 20-21: This sentence should be broken-up so it's easier to read. It took me at least 2 attempts to pause in the appropriate places and follow the entire sentence. I recommend something like: "Ice discharge is the product of the ice velocity and thickness across the grounding line. Satellite rate altimetry is used to retrieve information about ice surface velocity. Ice thicknesses are estimates from airborne radar or, in the absence of radar observations, using surface elevation observations under the assumption that the ice is floating."

p. 3, lines 6-7: Change to . . ."(GIA), which is the response of the lithosphere to changes in surface loading."

p. 3, line 10: Remove the comma before "by"

p. 3, lines 14-16: Either remove these lines "We combined our. . . to direct observations of ice surface height from ICESat." or rephrase. I don't think you need to go into much detail at this point and these two sentences are currently really difficult to follow.

p. 3, lines 18-20: I find this sentence confusing. I follow that you obtain similar estimates of dynamic mass change from GRACE and ICESat observations but I don't understand what you mean that they "can be used to model surface elevation changes that are comparable with altimetry observations". ICESat data are altimetry observations. Do you mean that you can use GRACE data to estimate the surface elevation change expected due to dynamic change using the methods you describe here? If so, you need to revise the sentence so that is clear.

p. 3, line 22 to p. 4, line 6: You start off by stating that there is a positive mass change trend across the study region but then go on to say the region is roughly in balance. Please revise to present a more consistent background on mass change estimates from the region.

p. 4, lines 18-26: I assume that the "slope correction" you present in Equation 1 is

an effort to account for drifting snow across the ice sheet surface. I believe this is already accounted for in RACMO (as you state in the appendix) so you may be "double-counting" for snowdrift. If you are referring to some other mechanism, please make that more clear. Additionally, the last sentence here should state specifically where you obtain estimates for these variables, not just what is "typical".

p. 5, line 1: Replace "seen" with "measured"

p. 5, line 2: Remove comma before "as well as the effect of GIA"

p. 6, lines 5-6: Replace with "The solutions to Equations 4 and 5 are the change in ice mass, DM/dt, and surface elevation, dH/dt, associated with changes in ice dynamics", with the proper subscripts and superscripts added.

p. 6, line 6: Replace "mass rate and height rate" with "rate of change in mass and surface elevation"

p. 6, lines 17-19: There's an assumption inherent in these conversions that the entire ice sheet thickness is composed of glacier ice when we know this is not the case. It would be helpful to have an estimate provided somewhere of the fraction of the total thickness that is firn versus ice. If the firn column is only ∼50m but the ice is ∼2000m thick, this assumption is fairly reasonable. However, if the ice is relatively thin and/or the firn column is very thick, then the density used for these conversations should be reduced.

p. 6, line 20 to p. 7, line 4: Shouldn't you be adding dH/dt estimate from GRACE to the dH/dt estimates for SMB and firn AND dH/dt from GIA to get a signal that is equivalent to the ICESat dHdt? Also, if dH/dt of ice from GRACE and ICESat are not equal, the discrepancy could also be caused by the spatial and temporal variations in the density of the ice used in the conversion, inability of the SMB and/or firn model to realistically simulate surface changes, in addition to errors/limitations in data processing techniques. You should list all potential sources of error briefly here.

p. 7, lines 9-10: Split the sentence so that it reads "...measured by GRACE. Figure 2a shows the map of the GRACE mass change signal and Figure 2b shows a time series for a coastal location near..."

p. 7, lines 17-19: Why not use the ICE_5G_C results? Are the other results more realistic/better for some reason?

p. 7, line 20. Break into two sentences so that you now have "... snowfall and ice discharge. The GIA-corrected GRACE mass change data suggest a positive mass trend of $\sim$32 +/- 8 mm w.e. yrˆ-1 between 30°E and 70°E and a substantial increase in mass from 2003-2009 (Fig. 3b)." The anomaly you list should be averaged over this entire region. I think the anomaly is only estimated over a smaller region currently, which is a bit misleading. Also, how can you attribute this to SMB? The SMB signal is actually from RACMO, correct? Are you presenting the mass gain estimated by RAMCO for SMB only or the GRACE SMB+discharge mass signal?

p. 8, line 3: How do you convert the ICESat data into spherical harmonics? What precisely does this mean? Does it mean you spatially average the data in some way? Please elaborate.

p. 8, lines 5-17: I find this section to be really difficult to follow. You should make it clear that you are using the RACMO models to estimate SMB contributions to the GRACE and ICESat signals. Saying "For both RAMCO2 models the ice dynamic estimates" and "Using RAMCO2.3 the ice dynamic estimates" reads a bit like you are estimating the dynamic signal directly from RACMO. Are the rates of dynamic mass change averages over the entire time period for each observational platform? Are the RMSE estimates the RMSE of the difference in SMB between the two RACMO versions over the entire study region? It would be helpful to have numeric estimates clearly presented in this section along with their error estimates. It would also be helpful to focus on just the difference in RACMO SMB over the study region, with discussion as to which version produces more realistic results when used to tease-out the dynamic signal, then

compare the dynamic change estimates. Right now there's just too much going on at once.

p. 8, lines 18-26: As mentioned earlier, I think you need to add dH/dt from GIA into your GRACE-derived dH/dt estimates. You should also include values for the trends you discuss here so that the reader can discern "strong" and "weak" trends.

p. 9, lines 12-16: You should include maps of uncertainty with the dynamic change estimates in the text (move Figure A2 to the body of the manuscript).

p. 10, line 1: Can you substantiate this remark that the different GIA models have a small effect on the ice dynamic change estimates? There have been rather large error bars in previous Antarctic mass change estimates from GRACE that have been largely attributed to uncertainty in the GIA signal.

p. 10, line 3: Replace "We believe" with "Our data suggest"

p. 10, line 5: Is the different statistically significant?

p. 10, lines 12-13: Replace with "Thus, a comparison of estimated changes in ice dynamics derived from GRACE and altimetry observations not only provides information about dynamic mass change, but may also help to identify regions where models fail to accurately simulate variations in SMB."

p. 12, line 1: Remove comma after "grid"

p. 16: What about uncertainties associated with GIA? I expect that these are quite large but they are seemingly overlooked. They are likely difficult to quantify but you could likely obtain uncertainty estimates computed for each GIA model from the model developers.

Figure 2: Include the name of the model used in the timeseries of GIA.

Figure 3: Remove the last sentence in the caption.

Figure 4: The dynamic mass change rates are obtained from GRACE and ICESat using RACMO to parse-out the SMB signal, not from RACMO using GRACE and ICESat as currently presented.

Please also note the supplement to this comment:
http://www.the-cryosphere-discuss.net/tc-2016-269/tc-2016-269-RC2-supplement.pdf

---

## Author Comment (AC1) · 2 Mar 2017

General Comment

Kallenberg et al report on a new approach for estimating ice dynamic rates for mass balance assessment from EO data and apply this for a study site in East Antarctica where an increase in ice mass has been observed in recent years. In the approach the ice dynamic are estimated by combining modelled SMB rates with gravity observations from GRACE and laser altimetry observations from ICESat. The derived IDR is combined with modelled elevation changes due to snow processes for comparison with

measured elevation changes from ICESat. The authors find the estimated ice dynamic rates from GRACE and ICESat of similar magnitude and modelled elevation changes in correlation with direct altimetry observations. This is a well written, illustrated an referenced methodological manuscript and a valuable and original contribution for the glaciology community. I would suggest a few minor corrections/clarifications to help improve the manuscript.

Specific Comments

Pg2 – Ln 20-23: There appears to be a mix up here. Ice velocity is derived from satellite radar interferometry and related methods in the mentioned studies, not from altimetry.

Changed on Pg2, Ln24.

—

Pg5 – Ln 12-17: Studies have shown that leakage from oceanic geophysical signals may bias mass rate estimates for Antarctica from GRACE significantly. How is this quantified or dealt with in the approach?

We converted the altimetry observations into spherical harmonics so that we represent the ice height information with the same spatial resolution as the mass change information. By doing this we impose the same potential leakage on to the altimetry observations. We added a comment to this effect on Pg.13, Ln.18-20.

—

Pg6 – Ln12: trend due to SMB

Changed on Pg11, Ln 25.

—

Pg7 – Ln24: Eq. 7 & Eq. 9 do not exist, I assume Eq. 4 & Eq. 5 are meant

This has changed by incorporating the appendix into the body of the manuscript. The

new equation numbers have been adapted.

—

Pg13 – Ln4: Ligtenberg et al.

Changed on Pg 7, Ln8.

—

Pg13 – Ln18: Eq. A3

This has changed by incorporating the appendix into the body of the manuscript. The new equation numbers have been adapted.

—

Pg15 – Ln5&Ln11: Equation (A7)

This has changed by incorporating the appendix into the body of the manuscript. The new equation numbers have been adapted.

—

Pg15 – Ln25: (Appendix A2)

The appendix has been deleted.

—

Pg16 – Ln14: I assume Eq. 1 is meant here

This has changed by incorporating the appendix into the body of the manuscript. The new equation numbers have been adapted.

—

Pg16 – Ln15: I assume Eq. A3 is meant here

This has changed by incorporating the appendix into the body of the manuscript. The new equation numbers have been adapted.

—

Fig. 1: Invert color scale

The plot is of the velocity of ice. The colour scheme is therefore somewhat arbitrary. We reversed the colour scheme but it turns the figure into something very dark, with the background colour over the region being dark blue rather than white. We prefer our original colour scheme and so have chosen to leave it as it was.

—

Fig. 2: What is the white cross in Fig. 2a

Added, now Figure 3, Pg26.

—

Fig. 2: Y-axis label: dM/dt in mm or mm/yr?

The Y-axis label has been updated, Fig. 3b, Pg.24.

—

Fig. 3: (m yr-1): please check scale here

Corrected, now Fig.4, Pg.25

—

Anonymous Referee #2

Summary: The manuscript back-calculates dynamic mass loss from East Antarctica using repeat GRACE and satellite altimetry observations paired with models of surface mass balance, firn compaction, and glacial isostatic adjustments. The authors find that rates of dynamic mass change inferred from the two different satellite observational platforms yield similar results, indicating that either platform can provide reasonable estimates of dynamic mass change when paired with current model outputs. The authors also infer that the good agreement between dynamic mass change estimates derived from the two different satellite platforms indicates that the most up-to-date RACMO model provides accurate estimates of surface mass balance for East Antarctica. The results of the paper are interesting in that they show models of SMB, firn compaction, and GIA are accurate enough to allow us to tease-out the ice dynamics signal from repeat gravity and laser altimetry observations. There are several relatively small modifications that would improve the overall quality of the manuscript, including: 1) removal/adjustment of commas (described below), which often break sentences into somewhat awkward fragments, 2) incorporation of data and methods description that is currently contained in the appendices, and 3) change references to "ice dynamics rates" to "mass change rates due to dynamic change" or something similar My biggest concern is in regard to (2) above. I highly recommend that more information on the datasets and methods is included in the body text of the manuscript. The details regarding how (1) you process the GRACE and ICESat data, (2) the versions of RACMO used in the study differ from each other, and (3) the firn compaction converts SMB input to estimates of surface elevation change are incredibly important for assessing the validity of the results and for reproducing the method elsewhere. Although this will lengthen the manuscript, I think that adding more detail regarding the data used in the analysis is imperative. Given that the firn layer in Antarctica can be tens of meters thick and that the interpretation of altimetry data is incredibly sensitive to the accuracy of the firn compaction model (see Zwally et al., Journal of Glaciology, 2015 for an example) this is particularly important for the firn compaction model. As a follow-up, why use RACMO2.1 to run the firn compaction model when RACMO2.3 was found to produce better results for the GRACE-ICESat dynamic change comparison? This is inconsistent.

1) remove/adjust commas

Changes performed as mentioned in detailed comments.

—

2) move data and method description from appendices into text

The appendices were incorporated into the body of the manuscript.

—

3) change "ice dynamic rates" to "mass change rates due to ice dynamic changes" or similar.

Changed throughout the manuscript.

—

4) Why use RACMO2.1 for the firn compaction model if RACMO2.3 was found to estimate SMB more accurately.

As explained within the firn compaction section the model has been tuned to fit available observations. Using RACMO2.3 means we would have to fine-tune the model differently in order to adjust it to available observations, due to the differences in the SMB. This is already explained in the manuscript and we didn't add anything further.

—

Detailed Comments:

p. 2, line 4: Change to "and bedrock uplift rates."

Changed on Pg2, Ln4.

—

p. 2, line 5: Move the comma after "ice lost" forward so it is after "balance"

Changed on Pg2, Ln5.

[Figure]

—

p. 2, line 11: Either move the definition of firn so that it comes earlier in the sentence or remove it entirely. Currently it's in an awkward location.

Changed on Pg2, Ln14.

—

p. 2, line 14: Change to "results in a change in the ice sheet surface elevation without..."

Changed on Pg2, Ln15.

—

p. 2, line 15: Remove "potentially"

Removed on Pg2, Ln17.

—

p. 2, lines 20 -21: This sentence should be broken - up so it's easier to read. It took me at least 2 attempts to pause in the appropriate places and follow the entire sentence. I recommend something like: "Ice discharge is the product of the ice velocity and thickness across the grounding line. Satellite rate altimetry is used to retrieve information about ice surface velocity. Ice thicknesses are estimates from airborne radar or, in the absence of radar observations, using surface elevation observations under the assumption that the ice is floating."

Changed and rewritten on Pg2, Ln 23-26.

—

p. 3, lines 6-7: Change to..."(GIA), which is the response of the lithosphere to changes in surface loading."

Changed on Pg3, Ln10.

—

p. 3, line 10: Remove the comma before "by"

Removed on Pg3, Ln14.

—

p. 3, lines 14-16: Either remove these lines "We combined our... to direct observations of ice surface height from ICESat." or rephrase. I don't think you need to go into much detail at this point and these two sentences are currently really difficult to follow.

Changed on Pg3, Ln 18-21.

—

p. 3, lines 18-20: I find this sentence confusing. I follow that you obtain similar estimates of dynamic mass change from GRACE and ICESat observations but I don't understand what you mean that they "can be used to model surface elevation changes that are comparable with altimetry observations". ICESat data are altimetry observations. Do you mean that you can use GRACE data to estimate the surface elevation change expected due to dynamic change using the methods you describe here? If so, you need to revise the sentence so that is clear.

Changed on Pg3, Ln 18-21.

—

p. 3, line 22 to p. 4, line 6: You start off by stating that there is a positive mass change trend across the study region but then go on to say the region is roughly in balance. Please revise to present a more consistent background on mass change estimates from the region.

We have restructured and rephrased the sentences to explain the mass balance situation more clearly for our study region. Changed Pg3, Ln 25 to Pg4, Ln 9.

p. 4, lines 18-26: I assume that the "slope correction" you present in Equation 1 is an effort to account for drifting snow across the ice sheet surface. I believe this is already accounted for in RACMO (as you state in the appendix) so you may be "double-counting" for snowdrift. If you are referring to some other mechanism, please make that more clear. Additionally, the last sentence here should state specifically where you obtain estimates for these variables, not just what is "typical".

The "slope correction" introduced by Helsen et al. (2008) (supporting material, page 2) is applied to the parameterisation of Kaspers et al. (2004) to estimate surface snow densities. Their relation was found to obtain more realistic values that are in better agreement with available density observations. It is still used by Ligtenberg et al. (2011) to estimate their surface snow densities for the firn compaction modelling. Neither from the Helsen et al. (2008) nor the Ligtenberg et al. (2011) publication it is stated that the "slope correction" is applied to account for drifting snow but rather to fine-tune the surface snow parameterisation. We reworded this part when incorporating the appendix into the manuscript: Pg7, Ln23 – Pg.8, Ln3.

—

p. 5, line 1: Replace "seen" with "measured"

Changed on Pg10, Ln12.

—

p. 5, line 2: Remove comma before "as well as the effect of GIA"

Removed on Pg10, Ln13.

—

p. 6, lines 5-6: Replace with "The solutions to Equations 4 and 5 are the change in ice mass, DM/dt, and surface elevation, dH/dt, associated with changes in ice dynamics",

with the proper subscripts and superscripts added.

Changed on Pg11, Ln 15.

—

p. 6, line 6: Replace "mass rate and height rate" with "rate of change in mass and surface elevation"

Changed on Pg11, Ln19.

—

p. 6, lines 17-19: There's an assumption inherent in these conversions that the entire ice sheet thickness is composed of glacier ice when we know this is not the case. It would be helpful to have an estimate provided somewhere of the fraction of the total thickness that is firn versus ice. If the firn column is only ∼50m but the ice is ∼2000m thick, this assumption is fairly reasonable. However, if the ice is relatively thin and/or the firn column is very thick, then the density used for these conversations should be reduced.

Our assumption is that a mass change observed by GRACE is caused by a change in SMB, GIA and ice dynamic. As firn compaction has no effect on GRACE observations, any changes within the firn layer are covered by the rate of change due to SMB, which occurs within the firn layer. We estimate the rate of change due to SMB and GIA, and assume that the remaining signal belongs solely to changes in the glacier ice, specified by ice dynamic signal. Elevation variations detected by altimetry are caused by changes in SMB, firn compaction, GIA and ice dynamic. We estimate the rate of change due to SMB, firn compaction and GIA, and assume that the remaining signal belongs solely to changes in the glacier ice, specified by ice dynamic signal. As we assume that the remaining signal is solely due to changes within the glacier ice, we have chosen to use the density of glacier ice. It is true that ice dynamic changes affect the entire ice sheet (ice and firn column) but here we address both columns separate

from each other, and changes within the firn layer (SMB+firn compaction) have been considered separately.

We added a brief explanation of this at Pg. 11 Ln.16-19.

—

p. 6, line 20 to p. 7, line 4: Shouldn't you be adding dH/dt estimate from GRACE to the dH/dt estimates for SMB and firn AND dH/dt from GIA to get a signal that is equivalent to the ICESat dHdt? Also, if dH/dt of ice from GRACE and ICESat are not equal, the discrepancy could also be caused by the spatial and temporal variations in the density of the ice used in the conversion, inability of the SMB and/or firn model to realistically simulate surface changes, in addition to errors/limitations in data processing techniques. You should list all potential sources of error briefly here.

This is correct, changes due to GIA should be included. Initially we left it out of the equation, because the satellite data used has already been corrected for GIA. We added it to Eq.13 on Pg12.. List of potential error sources added on Pg12, Ln15-17.

—

p. 7, lines 9-10: Split the sentence so that it reads "...measured by GRACE. Figure 2a shows the map of the GRACE mass change signal and Figure 2b shows a time series for a coastal location near..."

Changed on Pg12, Ln22.

—

p. 7, lines 17-19: Why not use the ICE_5G_C results? Are the other results more realistic/better for some reason?

The ICE_6G_C model is an updated version of the ICE_5G_C and provides a better estimate on GIA uplift rates according to Peltier et al. (2015) and Argus et al. (2014). We therefore chose to use only their most recent model.

—

p. 7, line 20. Break into two sentences so that you now have "... snowfall and ice discharge. The GIA-corrected GRACE mass change data suggest a positive mass trend of ∼32 +/-8 mm w.e. yrˆ-1 between 30°E and 70°E and a substantial increase in mass from 2003-2009 (Fig. 3b)." The anomaly you list should be averaged over this entire region. I think the anomaly is only estimated over a smaller region currently, which is a bit misleading. Also, how can you attribute this to SMB? The SMB signal is actually from RACMO, correct? Are you presenting the mass gain estimated by RAMCO for SMB only or the GRACE SMB+discharge mass signal?

The value of the mentioned anomaly was pointing out the two regions with the strongest observed anomaly, which was stated in the original sentence "with a positive anomaly of ∼32+/-8 mm w.e. yrˆ-1 near 40°E and 55°E". We did break the sentence down and removed the stated anomaly for a better understanding. It is also correct that the trend should not be attributed to SMB, it should have said "mass" instead of "SMB". This has been changed.

Changed on Pg13, Ln12-13.

—

p. 8, line 3: How do you convert the ICESat data into spherical harmonics? What precisely does this mean? Does it mean you spatially average the data in some way? Please elaborate.

The values of the ice dynamic estimates have been converted into the C and S co-efficients of spherical harmonics. This allowed us to convert our spatial grid into a spherical harmonic grid. This is a basic conversion and we don't feel it necessary to explain it in detail within the manuscript.

—

p. 8, lines 5-17: I find this section to be really difficult to follow. You should make it clear

that you are using the RACMO models to estimate SMB contributions to the GRACE and ICESat signals. Saying "For both RAMCO2 models the ice dynamic estimates" and "Using RAMCO2.3 the ice dynamic estimates" reads a bit like you are estimating the dynamic signal directly from RACMO. Are the rates of dynamic mass change averages over the entire time period for each observational platform? Are the RMSE estimates the RMSE of the difference in SMB between the two RACMO versions over the entire study region? It would be helpful to have numeric estimates clearly presented in this section along with their error estimates. It would also be helpful to focus on just the difference in RACMO SMB over the study region, with discussion as to which version produces more realistic results when used to tease-out the dynamic signal, then compare the dynamic change estimates. Right now there's just too much going on at once.

We have modified the entire section to improve our discussion. We clarified that we use the modelled SMB estimates using RACMO: Pg13, Ln23-27. We added that the mentioned rates of change provide the information that the obtained rates are between the two values within a certain region: Pg13, Ln26-27, and that the provided RMSE estimates are averaged over the study area: Pg14, Ln 6.

—

p. 8, lines 18-26: As mentioned earlier, I think you need to add dH/dt from GIA into your GRACE-derived dH/dt estimates. You should also include values for the trends you discuss here so that the reader can discern "strong" and "weak" trends.

Initially we included values for the discussed trends. However, following the discussion including values for the trends reads somewhat awkward, which was the reason for us to exclude specific values. The reason why we find this not suitable is that we compare the regions in where the positive and negative trends have been modelled and correlate with the ICESat observations, rather than presenting the actual values of the trends. As stated in the manuscript the values don't correlate precisely with each

other. This makes it hard to follow actual trend-results when reading the discussion and comparing the figures. As we find that including values for the trends is not fitting we have therefore not added any values.

dH/dt from GIA added on Pg14, Ln10.

—

p. 9, lines 12-16: You should include maps of uncertainty with the dynamic change estimates in the text (move Figure A2 to the body of the manuscript).

Uncertainties and figures added to main text on Pg15-16, and P. XY, respectively.

—

p. 10, line 1: Can you substantiate this remark that the different GIA models have a small effect on the ice dynamic change estimates? There have been rather large error bars in previous Antarctic mass change estimates from GRACE that have been largely attributed to uncertainty in the GIA signal.

Estimated uplift rates due to GIA in our study region are between ∼0-1mm (W12a, Whitehouse et al., 2012) and ∼0-3mm (ICE-6G_C, Peltier et al., 2015). While the effects are larger on the GRACE observations than the altimetry observations it still remains very small. Consequently, even a 100% error in the modelled GIA values would have only a very small impact on the results shown.

We added a comment to this effect on Pg.13, Ln.10 and Pg.16, Ln1-5.

—

p. 10, line 3: Replace "We believe" with "Our data suggest"

Changed on Pg16, Ln25.

—

p. 10, line 5: Is the different statistically significant?

Yes, the difference is statistically significant. This was added on Pg. 16, Ln27 to Pg. 17, Ln2.

—

p. 10, lines 12-13: Replace with "Thus, a comparison of estimated changes in ice dynamics derived from GRACE and altimetry observations not only provides information about dynamic mass change, but may also help to identify regions where models fail to accurately simulate variations in SMB."

Changed on Pg17, Ln 8.

—

p. 12, line 1: Remove comma after "grid"

Removed, now on Pg6, Ln4.

—

p. 16: What about uncertainties associated with GIA? I expect that these are quite large but they are seemingly overlooked. They are likely difficult to quantify but you could likely obtain uncertainty estimates computed for each GIA model from the model developers.

Uncertainties for GIA models are not provided with the models. GIA models are based on available observations and assumptions about the Earth's viscosity profile and ice history. They are tuned to fit by performing a parameter search for the best-fitting ice sheet history and earth rheology values. Velicogna and Wahr (2006) estimated an uncertainty by defining a lower bound and an upper bound, using the minimum and maximum trend of a GIA model. Their uncertainty corresponds to the bounds of the GIA range, while their best estimate on GIA uplift rates is the midpoint of this range. Nevertheless, across our study region GIA uplift rates are small and differences between the models are < 2 mm/yr. Therefore, the error in the modelled GIA signal in

our study region are likely to be small.

We added a comment on Pg.16 Ln.1-5.

—

Figure 2: Include the name of the model used in the time series of GIA.

The viscoelastic uplift shown in the figure is derived from GRACE observations using the empirical formula that was shown by Purcell et al. (2011) to be valid for all realistic ice/earth models where there has been no change in load for the past 5000 years. That is, the result is not related to any particular GIA model at all. This is explained in Section 3.1 on Pg.5, Ln.14-17.

—

Figure 3: Remove the last sentence in the caption.

Removed, now Figure 4, Pg.27

—

Figure 4: The dynamic mass change rates are obtained from GRACE and ICESat using RACMO to parse-out the SMB signal.

Changed, now Figure 5, Pg. 28.

Please also note the supplement to this comment:
http://www.the-cryosphere-discuss.net/tc-2016-269/tc-2016-269-AC1-supplement.pdf

**Supplement:**

[revised manuscript text omitted]

(Fig. 2b)

| **Page 13: [1] Deleted** | **Bianca Kallenberg** | **28/02/17 2:24 PM** |

| **Page 13: [1] Deleted** | **Bianca Kallenberg** | **28/02/17 2:24 PM** |

3a

| **Page 13: [1] Deleted** | **Bianca Kallenberg** | **28/02/17 2:24 PM** |

3b

| **Page 13: [1] Deleted** | **Bianca Kallenberg** | **28/02/17 2:24 PM** |

3c

| **Page 13: [1] Deleted** | **Bianca Kallenberg** | **28/02/17 3:03 PM** |

rates

| **Page 13: [1] Deleted** | **Bianca Kallenberg** | **28/02/17 2:25 PM** |

,

| **Page 13: [1] Deleted** | **Bianca Kallenberg** | **28/02/17 2:25 PM** |

revealing a positive trend in SMB

| **Page 13: [1] Deleted** | **Bianca Kallenberg** | **28/02/17 2:26 PM** |

,

| **Page 13: [1] Deleted** | **Bianca Kallenberg** | **28/02/17 2:26 PM** |

with a positive anomaly of $\sim32 \pm 8$ mm w.e. yr$^{-1}$ near 40°E and 55°E, showing a

significant

| **Page 13: [1] Deleted** | **Bianca Kallenberg** | **28/02/17 2:25 PM** |

3b

| **Page 13: [2] Deleted** | **Bianca Kallenberg** | **28/02/17 2:26 PM** |

| **Page 13: [2] Deleted** | **Bianca Kallenberg** | **28/02/17 2:26 PM** |

| **Page 13: [2] Deleted** | **Bianca Kallenberg** | **28/02/17 3:03 PM** |

rates

| Page 13: [2] Deleted | Bianca Kallenberg | 28/02/17 2:27 PM |

| Page 13: [2] Deleted | Bianca Kallenberg | 28/02/17 3:04 PM |

rate estimates

| Page 13: [2] Deleted | Bianca Kallenberg | 28/02/17 3:11 PM |

| Page 13: [3] Deleted | Bianca Kallenberg | 28/02/17 2:30 PM |

For both RACMO2 models the ice dynamic estimates are of somewhat similar rate for the two estimates obtained from GRACE and ICESat, with the greatest ice dynamic rates obtained between 30°E and 50°E.

| Page 13: [3] Deleted | Bianca Kallenberg | 28/02/17 2:31 PM |

an

| Page 13: [3] Deleted | Bianca Kallenberg | 28/02/17 2:31 PM |

of

| Page 13: [3] Deleted | Bianca Kallenberg | 28/02/17 2:31 PM |

.

| Page 13: [3] Deleted | Bianca Kallenberg | 28/02/17 3:04 PM |

rates

[revised manuscript text omitted]

---

## Author Response (AR2)

**Summary:**
**The manuscript is greatly improved from its original version. Minor comments are listed below.**

**My only major comment is that you still must provide justification in the manuscript regarding why you apply RACMO2.1 data as input for your firn compaction model, not RACMO2.3. In your response you state that it is because you would need to reinitialize your firn compaction model but isn't that worthwhile if RACMO2.3 is more accurate? You need to justify why you use an older version of a model, even if it's simply because you don't have the computational resources to re-run the firn compaction model. This additional justification is particularly important because you compare dynamic mass change results obtained using SMB estimates from both versions of RACMO, despite using only one version of RACMO to estimate firn compaction.**

We added a more detailed statement on page 9-10, line 25-5, to clarify that it is not important for the outcome of our findings.

**You must also provide some context regarding what constitutes "strong", "weak", "slight", and "general" trends and "slight" over- or under-estimates in your discussion section even if you do not choose to include numbers in this section. Otherwise the reader cannot fully assess the validity of your results.**

Values added to the discussion on page 14, lines 6-17.

**Detailed Comments:**
**- p 1, line 18: Change to "… ice sheet mass balance and surface elevation, or to develop…"**

Changed on page 1, line 18.

**- p 1, line 24: "correlated"**

Changed on page 1, line 24.

**- p 2, line 5: Remove "dynamic" because it has a different connotation elsewhere in the manuscript**

Changed on page 2, line 5.

**- p 2, lines 14-15: Remove these lines about firn because you just included them in lines 10-11.**

Removed.

**- p 2, line 28: "are either estimated using satellite altimetry observations"**

Changed on page 2, line 26.

**- p 3, line 15: Change "ice sheet changes due to ice dynamic rates" with "ice sheet dynamic mass change"**

Changed on page 3, line 13.

**- p 5, line 21: "correct for cross-track variations in surface elevation due to surface slope"**

Changed on page 5, line 20.

**- p 5, lines 22-24: This is repetitive. You say the same thing at the beginning of the paragraph.**

Deleted.

**- p 6, line 7: "crossover height rate"? Do you mean rate of elevation change at track crossovers? You should say "height rate of change" or something similar throughout this section because "height rate" is somewhat ambiguous.**

Changed on page 6, line 4 and 5.

**- p 6, line 13: How does the inclusion of along- and cross-track elevation change estimates improve the accuracy of the slope correction? Aren't you using the same DEM for all your slope corrections? I would assume that the inclusion of elevation change estimates using both methods simply improves the spatial coverage of the results if you aren't using the data to construct a time-evolving DEM.**

The method developed by Hoffmann (2016) doesn't actually use an external DEM, rather it derives small patches of DEMs at each cross-over region. We have modified several sentences in section 2.3 to clarify this issue: pg 6, Ln.4-11

**- p 7, line 1: Replace ", evaluated using" with "which is in good agreement with".**

Changed on page 7, line 25.

**- p 9, line 24: Replace "have been obtained" with "were obtained".**

Changed on page 9, line 23.

**- p 15, line3: Replace "around 8 mm" with "~8 mm".**

Changed on page 15, line 2.

[revised manuscript text omitted]
 **(DEM) for each crossover region** directly from the data. **The DEM is then** used to estimate the **cross-track** slope at **the** crossovers (Hoffman, 2016).

The slope estimates at the crossovers are then interpolated **along-track** to remove the **cross-track** slope from the along-track measurements. Although the elevation change estimates from along-track

10 measurements are naturally less precise than the rate estimates at crossovers, combining both methods significantly increases the accuracy of the **cross-track** slope correction **applied to the** along-track **data** (Hoffman, 2016).

**3.3. RACMO2/ANT**

[revised manuscript text omitted]